# Effects of a Modified Tap Dance Program on Ankle Function and Postural Control in Older Adults: A Randomized Controlled Trial

**DOI:** 10.3390/ijerph18126379

**Published:** 2021-06-12

**Authors:** Qianwen Wang, Yanan Zhao

**Affiliations:** School of Sports Science and Physical Education, Nanjing Normal University, Nanjing 210023, China; 161502014@njnu.edu.cn

**Keywords:** postural control, ankle function, tap dance, older adults

## Abstract

Older adults are at a high risk of falling due to age-related degradations in physical fitness. This study aimed to examine the effects of a modified tap dance program (MTD) on ankle function and postural control in older adults. Forty-four healthy older adults (mean age = 64.1 years, with 9 men) were recruited from local communities and were randomly divided into the MTD group and the control (CON) group. The MTD group received 12 weeks of MTD training 3 times per week for 30 min per session. Outcomes were measured using the five times sit-to-stand test (FTSST) for ankle strength, the universal goniometer for ankle range of motion, and the Footscan^®^ to trace the center of pressure. Results revealed significant improvements in FTSST in the MTD group (mean difference = 1.01), plantar flexion (left = 9.10, right = 10.0). In addition, the MTD group displayed significantly more improvements at midtest than the CON group in FTSST (mean difference = 1.51) and plantar flexion (mean difference: left = 6.10; right = 4.5). Therefore, the MTD can be an effective exercise program for ankle function improvement, but it has limited effects on improving postural control among healthy older adults.

## 1. Introduction

People of advanced age with decreased postural control abilities in daily life have reported higher falling risks [1]. According to the World Health Organization, falling is the second leading cause of accidental or unintentional injury deaths worldwide [2]. One out of three adults aged over 65 years experience falling per year [3]. Abnormal postural control has been identified as one of the critical risk factors of falls in older adults [4].

To date, various methods have been proposed to reduce fall risks in the older population, of which exercise is one of the most effective ways to achieve balance improvement and fall prevention [4]. The majority of current exercise interventions, such as balance training, Tai Chi, vibration exercise, and stair climbing, are conducted to improve postural control by aligning the body’s center of pressure (COP) to the base of support [5,6,7,8,9]. However, no consensus has been reached on the most effective exercise format. A systematic review found that resistance, coordination, and multi-component exercise formats were equally effective in postural control [10], while Low and his colleagues found that resistance or multi-component exercise interventions could not change any of the related postural parameters [11]. Similar results were also found in the study by Jehu and his colleagues, where general balance training showed limited effects in postural sway [5]. One of the underlying reasons for this could be the lack of specifically designed exercise programs targeting the postural control system [12].

The human body is like an inverted pendulum [5], and the ankle plays a vital role in maintaining body balance. The ankle strategy is always preferred during balance disturbance. It is elicited by the activation, in a distal to proximal recruitment pattern, of anterior muscles of the lower limbs and trunk to overcome an anterior perturbation [5]. Therefore, ankle strength and adequate ankle joint flexibility are essential for maintaining postural stability [13]. In addition, there is evidence that muscle strength in ankle dorsiflexion and plantarflexion can be used for fall prediction [14,15]. Given the aging-related degradations in physiological function among older adults, ankle muscle weakness and limited ankle flexibility become critical risk factors in postural stability and falling [5,13,16]. Therefore, exercise interventions targeting ankle function improvements would have potential effects on postural control and fall prevention. However, most of the existing feet and ankle exercise programs have mainly consisted of heel/toe lifts in sitting, standing, walking, or ankle rotation practice [17]. As suggested by the results from a meta-analysis study, a lack of adherence and insufficient training intensity because of formalistic exercises may be the underlying reason for the limited effects on ankle function, muscle strength, and functional abilities [17].

Dance is a kind of creative, appealing, and versatile activity. It includes an esthetic form of artistic expression which can contribute to emotional, cognitive, physical, and social integration [18]. Older adults who dance regularly show better performance in balance, flexibility, agility, and mental health [19,20]. Tap dance, an indigenous American art form, emphasizes ankle and foot movements. To date, limited evidence has been obtained regarding the effects of tap dance among older adults. Taking safety considerations into account, a modified tap dance program (MTD) based on the fitness characteristics of older adults was developed by our research team and has been published elsewhere [21]. A feasibility test has indicated that the MTD program is a safe and feasible exercise and can be applied to older adults’ daily life. In addition, a recent study found that a 16 week course of the MTD can reduce the plantar pressure of diabetic patients who are at risk of diabetic foot [22]. However, the effects of the MTD on ankle function and postural control among healthy older adults remained unknown. This study therefore aimed to examine the effects of the MTD on the improvement of ankle function and postural control in older adults. Results from this study can provide more knowledge on the benefits of ankle-targeted exercise formats and increase our knowledge regarding dance therapy and health promotion.

## 2. Materials and Methods

### 2.1. Aim

The purpose of this study was to examine the effects of an MTD program on improving ankle function and postural control in older adults.

### 2.2. Trial Design

This was a randomized, double-blinded, two-group parallel study conducted in local communities in Nanjing, China.

### 2.3. Participants

Participants were recruited through informational flyers distributed to three local community senior centers in a district. An initial pool of 61 healthy and independent older adults (aged between 60 and 74 years) was recruited. The qualified participants needed to be free from any limitations to exercise participation and to have had no falling history in the previous year. Participants were excluded if they had cognitive deficits assessed by the Chinese version of the Mini-Mental Status Examination (score < 24), self-reported vision or vestibular disorders, arthritis, or any other foot or ankle disease.

### 2.4. Intervention

The MTD group received 12 weeks of MTD training, with 3 sessions per week for 60 min per session. Each session included three phases: (1) 10 min of warm-up, (2) 45 min of MTD practice, and (3) 5 min of cooling down. The entire intervention protocol was conducted according to the classic three-stage learning theory (i.e., cognitive, associative, and autonomous stages) [23], and following the basic training principles (e.g., overload, progression, and reversibility). The training process was led by a senior certified dance instructor who was blinded to the aim of the study. The CON group received three health education lectures during the 12-week study period and were asked to keep their usual lifestyles without participating in any specific ankle or balance training.

### 2.5. Outcomes

Participants were included in the study and underwent three tests before (0 weeks), during (end of the 6th week), and immediately after the interventions (end of the 12th week). Testers were blinded to the intervention assignment and group allocation. The baseline data, including demographic and clinical characteristics of all the participants, were collected or tested at their first arrival.

#### 2.5.1. Ankle Function

Muscle strength: Ankle strength was evaluated using the five times sit-to-stand (FTSST) test. Participants were asked to perform five rises from a 43 cm-high chair as fast as possible with arms crossed over the chest. The total duration was recorded in seconds. To gain familiarity with the test procedure, each participant practiced twice before the formal test. The test–retest reliability of the FTSST is well accepted (ICC = 0.89, 95%CI = 0.79–0.95), and it has been used as a valid test for ankle muscle strength in dorsiflexion [24] and plantar flexion (R^2^ = 0.72) [25].Range of motion (ROM): A standard universal goniometer (XuBin Medical Equipment Company, Handan, China) was used to measure ankle ROM during active plantar flexion and dorsiflexion on both sides; i.e., left ankle dorsiflexion (LD-ROM), left ankle plantar flexion (LP-ROM), right ankle dorsiflexion (RD-ROM), and right ankle plantar flexion (RP-ROM). Participants were asked to sit on a treatment table with their knees fully extended (at 0°) and their feet hanging off the table. During the test, participants had to be completely relaxed, actively moving the ankle into dorsiflexion or plantar flexion from a neutral starting position until a firm end-feel was elicited. The goniometer axis was centered on the lateral malleolus, and the arms were aligned with the fibular shaft and the head of the fifth metatarsal. Each test was repeated twice, and the best score was recorded.

#### 2.5.2. Postural Control

The Footscan^®^ (RSscan, Footscan Balance, Version 7, Olen, Belgium) was applied to measure the sway of the COP in a stance. Participants were instructed to stand barefoot on the platform with hands touching their hips and eyes looking straight ahead or closed. Each participant was tested twice with 30 s rest intervals during the following three tests [26]: Test 1: standing with hands touching hips, feet together, and looking straight ahead; Test 2: standing with hands touching hips and eyes closed; Test 3: tandem stance (toes of the left foot touching the right heel) with hands touching hips, looking straight ahead. Each test lasted 33 s. The test was stopped if a participant’s foot left the floor.

### 2.6. Randomization

Participants who signed the consent form were randomly allocated to one of the two groups using the computer-generated random sequence. A student assistant who had not been involved in any other procedures of this study helped with participant allocation and informed participants of their group allocation by telephone.

### 2.7. Sample Size

The sample size was calculated using the G*Power analysis program (Version 3.1.9.2). Based on the results from a related study [27], a moderate effect size of 0.25 was adopted for sample size estimation. To ensure 80% probability that a treatment effect could be detected at a two-sided significance level of 0.05, a total of 35 participants was required. Considering a potential 20% dropout rate and additional variations, a sample size of 22 participants per group (44 participants total) was needed.

### 2.8. Data Analysis

Analyses were generated using SPSS version 24 (IBM Corp., Armonk, NY, USA). Demographic and clinical data are presented as means and SDs or as numbers and percentages. Based on the research purpose, a one-way repeated measures ANOVA was conducted in each group to examine the effects of time on test outcomes. Bonferroni corrections were applied to pairwise comparisons to control for type I errors. The one-way analysis of covariance (ANCOVA) taking the baseline value as the covariance was conducted at midtest and posttest to determine the group effects on each outcome parameter. The significance level was set at *p* < 0.05.

### 2.9. Ethical Approval

The experimental protocol was developed following the ethical guidelines of the Helsinki Declaration and was approved by the Human Ethics Committee of Nanjing Normal University (Code number: 202012001).

### 2.10. Trial Registration

The study protocol was registered with the Chinese Clinical Trial Registry on 30 January 2018 (Registration number: ChiCTR1800014714).

## 3. Results

A total of 44 participants with a mean age of 64.1 years qualified for this study. Women represented a large proportion in the study sample (nearly 80%), which is in line with the related studies. No adverse effects were reported. The relatively high attendance rate (average = 88.3%) coupled with the lack of drop-out during the intervention process indicates the feasibility of older adults integrating the MTD program into their daily life. Table 1 shows the demographic and clinical characteristics of the participants in each group. Figure 1 presents the participant flow.

### 3.1. Ankle Function

Results from the ANCOVA demonstrated significant group differences at midtest in FTSST, F (1, 41) = 5.50, *p* = 0.024, pη^2^ = 0.118; LP-ROM, F (1, 41) = 9.28, *p* = 0.004, pη^2^ = 0.184, and RP-ROM, F (1, 41) = 7.26, *p* = 0.01, pη^2^ = 0.15. No significant group differences were found in posttest (See Table 2).

The one-way repeated measures ANOVA displayed significant time effects on FTSTS, LP-ROM, and RP-ROM for both the MTD and the CON. In addition, all the significant time effects occurred between pretest and midtest and between pretest and posttest, while no significant differences were found between midtest and posttest (Table 2). The MTD, and not the CON, showed significant time effects in RD-ROM, with F (1.90, 40.4) = 3.55, *p* = 0.04, and pη^2^ = 0.144. Pairwise comparisons found significant decreases in RD-ROM in the posttest compared to those in the pretest (mean difference = 4.42, *p* = 0.02). For other test parameters, although no statistically significant group and time effects were revealed, the gradual declining trends over time indicated that the MTD group had better performance than the CON group with a longer training duration.

### 3.2. Postural Control

Results from the ANCOVA revealed no significant group effects in all the test indicators, except for the ellipse area of COP in both feet stances with eyes open at midtest, with F (1.41) = 6.20, *p* = 0.02, and pη^2^ = 0.13 (See Table 3). Compared to the CON group, the MTD group showed a significantly lower value in the ellipse area of the COP (mean difference = −4.56).

Results from the repeated measure ANOVA showed significant time effects in the total traveled distance of the COP when standing with eyes open in the MTD group, with F (1.88, 39.5 = 3.79), *p* < 0.05, and pη^2^ = 0.153. Pairwise comparisons revealed significant increases from pretest to midtest (mean difference = 22.0). In addition, an obvious decrease in the ellipse area during tandem stance with eyes open was noted for the MTD group (mean difference = 18.5). No significant time effects were observed in the CON group.

## 4. Discussion

This study examined the effects of a 12-week MTD practice on ankle function and postural control among a group of older community residents. Results revealed significant improvements in ankle strength and right and left plantar flexion but limited influences on postural sway during a double-foot stance or tandem stance.

Given the inverted clock structure of the human body, the ankle plays a vital role in the maintenance of body balance [28]. Older adults with a falling history have reported significant decreases in the muscle strength of both knees and ankles compared to their counterparts. In addition, muscle strength decreases faster in the ankle than in the knee [29]. All these factors emphasize the importance of ankle strength in the postural control of older adults. In the present study, significant improvements in FTSST were found after a 12-week exercise intervention, which is consistent with a previous study [30]. However, it was unexpected that significant group effects would be displayed in the midtest instead of the posttest, and similar results were also found in the plantar flexion. The non-significant group differences at posttest and the improvement in the control group can be partly explained by the “context effect” [31]. Participants receiving health-related workshops gain knowledge about healthy behaviors, and this knowledge promotes an active lifestyle to an extent. In addition, attending a test three times would be indicative for the participants in the control group and encourage participants from the control group to think about (and perhaps act upon) behaviors they could undertake to improve their test performance [32].

Aging-related degradations in ankle ROM are often reported in older adults [33]. Along with specific muscle strength, adequate ankle ROM allows for efficient force generation and ankle strategy execution during balance disturbance [23]. Contrary to those studies that found significant and positive training effects in plantar flexion and dorsiflexion [34], the present study only showed significant improvements in plantar flexion. Moreover, there was a significant reduction in the ROM of the right dorsiflexion. This is quite unexpected but can be explained by the characteristics of tap dancing. Participants receiving the MTD were required to repeat numerous movements by striking their tap shoes against the floor. Moreover, participants had to wear tap shoes with a heel height of 4 cm during the whole training process. All these factors would enhance their muscle strength and ROM for plantar flexion. Future studies are suggested to investigate the underlying mechanisms for the non-significant training effects of tap dance on dorsiflexion.

Regarding the changes in postural sway as tested by tracing the COP in different standing situations, no significant treatment effects were shown for all the test parameters. This is entirely unexpected and inconsistent with the related studies that examined the efficacy of dance in older adults [35,36,37]. One potential reason for this could be related to the health status of participants. Participants in this study were apparently healthy and free from any restrictions in exercising. They may not have been able to gain exercise benefits in postural control to the same extent as those who were suffering from Parkinson’s disease [35,38] or who were at risk of falling [39,40]. Alternatively, the other potential reason could be related to differences in postural control assessment (dynamic postural control assessments in previous studies vs. static postural control assessments in the current study). Further studies are therefore suggested to examine the effects of tap dance on both dynamic and static postural control considering participants’ characteristics.

### Limitations

This study had several limitations. First, although the FTSST is a reliable and valid test for ankle muscle strength, the FTSST cannot distinguish strength changes in dorsiflexion from those in plantar flexion. Highly precise assessments of ankle muscle strength are therefore suggested to be applied to related studies. Second, all the participants were recruited from local senior centers in the same district, which may increase treatment contamination between intervention and control groups. However, to minimize the potential bias, the participants in the CON group were informed that they could participate in the MTD after the study. Finally, all the participants in this study were healthy and without apparent balance or joint problems. This would limit the effects of ankle-targeted exercise interventions on postural control.

## 5. Conclusions

The 12-week MTD can be effective for improving ankle muscle strength and the flexibility of plantar flexion among community-dwelling older adults. A longer training period, however, would be warranted to examine the effects of the MTD on postural control.

## Figures and Tables

**Figure 1 ijerph-18-06379-f001:**
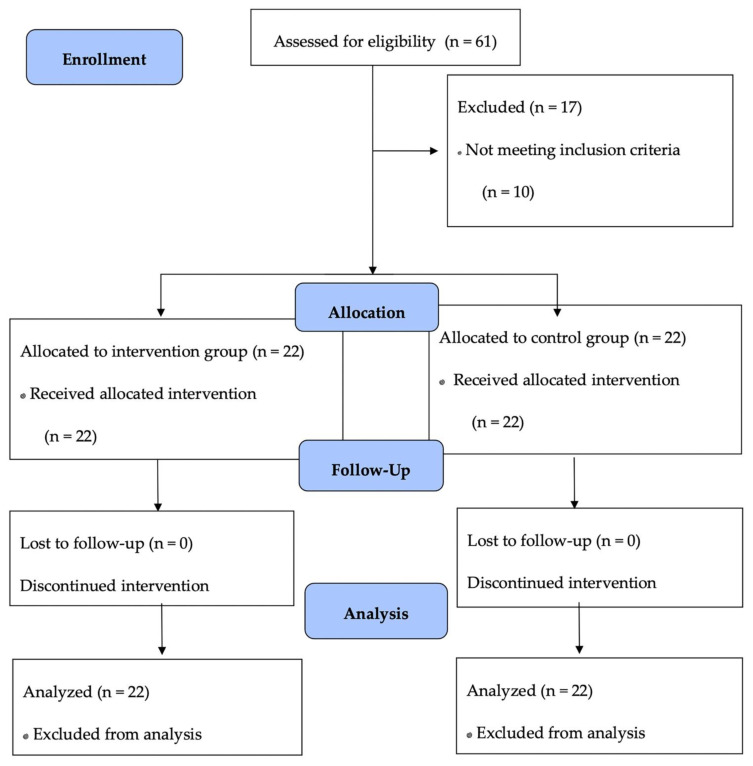
Participant flow.

**Table 1 ijerph-18-06379-t001:** Demographic and clinical characteristics of all the participants.

Parameters	MTD (*n* = 22)	CON (*n* = 22)	All (*n* = 44)
Age (years)	63.9 ± 4.21	64.6 ± 3.75	64.1 ± 4.02
Female (%)	18 (81.8%)	17 (77.3%)	35 (79.5%)
Height (cm)	158 ± 5.47	160 ± 13.6	160 ± 8.16
Body mass index (kg/m^2^)	24.0 ± 2.80	26.5 ± 6.5	25.2 ± 3.09
SBP (mmHg)	124 ± 13.6	126 ± 21.5	127.9 ± 14.5
DBP (mmHg)	75.9 ± 8.71	74.9 ± 8.99	74.5 ± 9.16

SBP, systolic blood pressure; DBP, diastolic blood pressure.

**Table 2 ijerph-18-06379-t002:** Between and within-group differences in ankle function-related parameters.

Tests	MTD (*n* = 22)	CON (*n* = 22)	Adjusted Difference (95% CI) *
Five times sit-to-stand test (s)			
Pretest	7.45 ± 1.51	8.49 ± 1.94	
Midtest	6.88 ± 1.06	8.39 ± 1.95	−1.51 (−1.81, −0.14) ^
Posttest	6.44 ± 1.28 ^#^	7.05 ± 1.19 ^#^	−0.61 (−0.98, 0.44)
Range of motion in left dorsiflexion (°)			
Pretest	14.2 ± 5.94	11.8 ± 5.62	
Midtest	13.2 ± 6.33	12.1 ± 7.74	1.10 (−3.72, 5.12)
Posttest	10.4 ± 7.05	10.9 ± 5.51	−0.50 (−4.45, 3.52)
Range of motion in left plantar flexion (°)			
Pretest	30.4 ± 5.96	30.2 ± 8.15	
Midtest	44.0 ± 4.83 ^#^	37.9 ± 8.27 ^#^	6.10 (2.04, 10.1) ^
Posttest	39.5 ± 7.22 ^#^	37.8 ± 5.88 ^#^	1.70 (−2.39, 5.67)
Range of motion in right dorsiflexion (°)			
Pretest	14.2 ± 6.91	14.5 ± 8.45	
Midtest	11.6 ± 5.81	12.5 ± 7.57	−0.90 (−4.98, 3.25)
Posttest	9.78 ± 6.02 ^#^	11.7 ± 5.75	−1.92 (−5.53, 1.73)
Range of motion in right plantar flexion (°)			
Pretest	29.8 ± 6.37	30.4 ± 10.9	
Midtest	42.3 ± 5.86 ^#^	37.8 ± 6.01 ^#^	4.50 (1.14, 8.00) ^
Posttest	39.8 ± 7.15 ^#^	38.2 ± 5.36 ^#^	1.60 (−2.19, 5.55)

* by ANCOVA adjusted for baseline value, *p* < 0.05; ^#^ significant differences compared with pretest; ^ significant differences between groups; MTD, modified tap dance program group; CON, control group.

**Table 3 ijerph-18-06379-t003:** Between and within-group differences in postural control-related parameters.

Tests	MTD	CON	Adjusted Difference 95% (CI) *
Test 1: Standing with both feet, eyes opened			
Total travelled distance of center of pressure (mm)			
Pretest	105 ± 26.8	154 ± 62.6	
Midtest	127 ± 43.5 ^#^	185 ± 68.0	−58.0 (−58.9, 7.08)
Posttest	125 ± 47.2	185 ± 120	−60.0 (−103, 20.9)
Ellipse area (mm^2^)			
Pretest	6.58 ± 5.13	8.23 ± 5.59	
Midtest	6.54 ± 4.05	11.1 ± 6.69	−4.56 (−7.38, −0.77) ^
Posttest	7.40 ± 5.49	8.24 ± 4.77	−0.84 (−2.83, 2.73)
Test 2: Standing with both feet, eyes closed			
Total travelled distance of center of pressure (mm)			
Pretest	124 ± 32.6	175 ± 85.3	
Midtest	138 ± 56.9	171 ± 59.1	−33.0 (−46.9, 22.6)
Posttest	137 ± 62.8	184 ± 65.9	−47.0 (−50.6, 18.2)
Ellipse area (mm^2^)			
Pretest	5.91 ± 4.26	7.48 ± 4.34	
Midtest	7.02 ± 5.09	9.85 ± 7.21	−2.83 (−5.62, 1.65)
Posttest	7.40 ± 6.21	9.56 ± 5.43	−2.16 (−3.92, 2.06)
Test 3: Tandem stance with right foot forward, eyes opened			
Total travelled distance of center of pressure (mm)			
Pretest	452 ± 223	526 ± 263	
Midtest	438 ± 153	487 ± 124	−49.0 (−121, 45.8)
Posttest	488 ± 149	490 ± 187	−2.00 (−91.1, 113)
Ellipse area (mm^2^)			
Pretest	31.5 ± 73.7	16.0 ± 13.8	
Midtest	21.3 ± 10.6	21.5 ± 14.5	1.70 (−6.79, 8.50)
Posttest	13.0 ± 11.9	15.7 ± 9.27	−2.70 (−8.52, 4.54)

* by ANCOVA adjusted for baseline value; *p* < 0.05, ^#^ significant differences compared with pretest.

## Data Availability

The data presented in this study are available on request from the corresponding author. The data are not publicly available due to the privacy.

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
