# Peer review of "Effects of a Modified Tap Dance Program on Ankle Function and Postural Control in Older Adults: A Randomized Controlled Trial"

_ijerph, 2021, doi:10.3390/ijerph18126379_

Round 1
Reviewer 1 Report
The topic is of interest in its current form. It is only necessary to make some minor revisions.
The authors presented a potential interesting, relevant and current study regarding the effects of a modified tap dance programme (MTD) on 10ankle functions and postural control in older adults. However, there are some basic shortcomings in the introduction section that must be addressed in order to improve the quality of the paper.
Introduction.
The introduction is easy to read, however did not extend existing knowledge on this topic (dance and fall). Please expand on this concept (dance) and how it is relevant to this paper. It should include more updated references regarding the evidence that support dancing for preventing losses on functional aspects related to falls, such as balance, gait ability and flexibility. See for example Bianco et al. (2014): Group fitness activities for the elderly: an innovative approach to reduce falls and injuries. Aging Clin Exp Res. 2014 Apr;26(2):147-52
Methods
Some important information appears to be presently omitted from the methods section. Have you tested the normality of the distributions in the variables considered? If yes, what kind of test have you used?
Discussion
Moreover, the first paragraph of the discussion should at least state which hypotheses were supported. Then the authors should follow with how their results compare with similar data, and what the authors results adds to the literature (different / unique aspects of the data). Several points are made in the discussion, but it is not clear to this reviewer how results from the current study are novel or add to the literature. The authors shortly discuss several possible explanations for the findings. Please expand the conclusion section.
Reviewer 2 Report
The article presents the results of research evaluating the effectiveness of a therapeutic program with the use of tap dancing on ankle joint function. The research was carried out on 44 people aged over 60, divided into two equal groups: the control and the intervention group.
The effectiveness of the tap program was assessed by performing the sit-to-stand test, range of motion in the ankle joint, and stabilographic tests. All studies were conducted before, during and after the 12-week treatment program. I believe that the article was not prepared carefully and cannot be published in its current form.
Critical comments to the article:
1.lack of information about the measuring device used to measure the range of motion in the ankle joint
2.table 2 lacks a description of the abbreviations used, e.g. what does LD-ROM, LP-ROM, RD-ROM and RP-ROM mean - I guess these are ranges of movements in the ankle joints
3.if the above-mentioned values ​​are ranges of motion in the ankle joint, the presented data, e.g. LD-ROM - pre-test - MTD group = 14.2 = / - 5.94 indicate that the measurement of the angles has been made with a precision to hundredths of a degree - I do not know measuring devices offering such high precision of measurements
- I have big reservations about the chapter discussion, in which only 10 references were used and the obtained results were not compared to the measurements of other authors. I believe that this is a major failure.
- I have reservations about data analysis and interpretation. By analyzing the data summarized in Tables 2 and 3, it can be concluded that the positive impact of the tap program was only noted in the sit-to-stand test. The length of the center of preasurre (TTW)
worsened in both groups studied, although the area of ​​the ellipse in the MTD group decreased after the pretest. However, it is worth noting that it is similar to the result of the control group. It is also worth noting that the EA value in the pretest for the MTD group was over 30 mm2 and was twice as high as in the CON group. In my opinion, the positive impact of the tap program on the stabilographic measurements is ambiguous.
- Analyzing the results of the ranges of motion in the ankle joint presented in Table 2, similar changes in the range of motion in the MTD and CON groups were noted. On this basis, no positive influence of the tap program on the range of motion can be stated. Moreover, the data presented in Figure 3 do not coincide with the data in Table 2.
- I believe that the final conclusions are not correctly worded (see notes 5 and 6).
Round 2
Reviewer 2 Report
I am glad that the authors of the article took into account most of the comments in my previous review. This version of the article is definitely better. I recommend the article for publication after some minor corrections:
- Please provide the name of the measuring device used for the measurement ankle ROM. The description: "A standard universal goniometer was used to measure ankle ROM" is insufficient (line 121-122).
- The data shown in Fig. 2 are duplicated in Table 2. I recommend deleting Fig. 2
- The data shown in Fig. 3 are duplicated in Table 3. I recommend deleting Fig. 3.
Author Response
Dear Editors and reviewers,
First of all, we would like to thank you for your positive and constructive comments on our work. We appreciate the time taken to offer comments and suggestions for improvement. We have added detailed information about the standard universal goniometer used for the range of motion (lines 114-115 of the revised manuscript with track changes; please refer to the attachments) and deleted Figure 2 and Figure 3 according to the reviewer’s suggestions. We hope that the resulting revised manuscript is closer to one that is acceptable for publication.
Best regards,
Qianwen Wang & Yanan Zhao
3/6/2021
